# Criteria for Assessing Exposure to Biomechanical Risk Factors: A Research-to-Practice Guide—Part 1: General Issues and Manual Material Handling

**DOI:** 10.3390/life14111398

**Published:** 2024-10-30

**Authors:** Francesca Graziosi, Roberta Bonfiglioli, Francesco Decataldo, Francesco Saverio Violante

**Affiliations:** 1Occupational Medicine Unit, Department of Medical and Surgical Sciences, Alma Mater Studiorum University of Bologna, 40138 Bologna, Italy; francesca.graziosi@unibo.it (F.G.); roberta.bonfiglioli@unibo.it (R.B.); francesco.violante@unibo.it (F.S.V.); 2Division of Occupational Medicine, IRCCS Azienda Ospedaliero-Universitaria di Bologna, 40138 Bologna, Italy

**Keywords:** biomechanical risk factors, occupational diseases, musculoskeletal disorders, ergonomics, occupational medicine

## Abstract

Musculoskeletal disorders are the most prevalent occupational health problem all over the world and are often related to biomechanical risk factors; to control these risk factors, several assessment methods (mostly observational) have been proposed in the past 40 years. An in-depth knowledge of each method to evaluate biomechanical risk factors is needed to effectively employ them in the field, together with a robust understanding of their effective predictive value and limitations. In Part 1, some general issues relevant to biomechanical risk assessment are discussed, and the method for assessing manual material handling after receiving more robust validation data is reviewed (Revised NIOSH Lifting Equation), together with a discussion about variability of tasks. Similarly, for the assessment of the biomechanical exposure of the upper limb, the TLV for Hand activity (ACGIH^®^) is presented in Part 2 of this guide, together with criteria to proportion risk assessment to the working duration in part-time jobs.

## 1. Introduction

Data from the European Foundation for the Improvement of Living and Working Conditions in Dublin (https://www.eurofound.europa.eu/surveys/european-working-conditions-surveys, accessed on 4 September 2024) show that the most frequently reported health problems by European workers are musculoskeletal and that they are related to exposure to biomechanical risk factors, as is the case in the rest of the world.

Biomechanical risk factors (or, for short, “biomechanical factors”) refer to forces applied to parts of the body, or developed by the body, to perform work or maintain a position: work tasks that may require the evaluation of these factors are the manual handling of (excessive) loads, manual work involving force, speed, and continuity of movements, and the maintenance of awkward postures for prolonged periods (kneeling, working with the hands above the head, and so on).

Adequate physical activity is fundamental for musculoskeletal system health: however, an excess load compared to acceptable parameters needs to be avoided, as it is potentially harmful. Epidemiological studies suggest that high levels of exposure to biomechanical factors, especially when combined, can lead to an increased risk of the onset of disorders of the musculoskeletal system [1].

In many circumstances, manual activities induce high mechanical load on body areas as the result of the combination and interaction of all the factors characterizing the task: for example, it is worth noting the influence exerted by the posture of a body segment on its ability to develop muscle force, as well as, in the case of the lumbosacral spine, on the forces to which the intervertebral disks are subjected. Less frequently, there are situations characterized by the almost exclusive presence of a single biomechanical factor which, individually, can significantly overload the body structures involved.

Over the years, several methods have been proposed to assess the risk associated with exposure to biomechanical factors: a number of these methods have been examined for scientific reliability and validity [2].

The best available methods, however, have been developed concerning a specific work task, while most activities involve the performance, in a work shift, of tasks that are generally different from each other [3].

In the absence of a “gold standard” for assessing biomechanical exposure, it is therefore important to always consider the following:
-The employed method influences the results that will be obtained and must therefore be motivated;-The results obtained should be considered as a general trend indication and, even when expressed in numerical values, they cannot be evaluated as for other areas of occupational exposure where it is possible to have a deeper knowledge of the relationship between the level of exposure and the effect on health (consider, for example, occupational noise exposure).

Therefore, the assessment of biomechanical risk requires the adoption of a methodological path capable, on one hand, of guaranteeing the need to examine the multiplicity of biomechanical factors that together can contribute to determining an overload in the different body areas, and, on the other hand, of satisfying the need to proceed, if indicated, to a more in-depth analysis of one or more factors.

Several methods for the assessment of biomechanical exposure have been proposed in the scientific literature with different levels of practical feasibility; this work aims to bridge the gap between research and practice, presenting a set of criteria based on the most valid and up-to-date scientific evidence. It also considers the need to use methods that do not require disproportionate technical, material, financial, and time resources. The selection process is based on literature review and expert consensus. Specifically, all the methods selected were published in peer-reviewed journals or recognized books in the occupational field and presented most data about repeatability and validity and a clear description written in the English language.

Finally, we aim to provide some practical recommendations to collect, manage, and analyze data.

### 1.1. Preliminary Analysis: Selection of a Representative Sample and Location for the Assessment

Preliminary to the technical analysis is the definition of the field of risk assessment, i.e., the set of activities that must be examined: in this regard, it is necessary to distinguish between organizations that have a single operational location and those that operate in multiple sites.

In organizations that have only one operational location, the preliminary analysis, conducted through the examination of available data on the work organization (completed by an inspection), aims to identify the activities that must be assessed for exposure to biomechanical factors. The activities carried out are grouped as far as possible into homogeneous groups by exposure to biomechanical factors to make the subsequent technical analysis efficient in identifying “similar exposure groups” of workers (with a procedure similar to that recommended, for example, by the EN 689 standard [4] for exposure to chemical agents)**.**

When organizations operate in multiple sites, the preliminary analysis aims to identify the activities that must be evaluated and whether the assessment should include all company sites or a representative subsample.

In multi-site organizations in which the same activities are carried out in different sites, there is usually a high degree of standardization (similar equipment, organization, ratio between workforce, and volume of activity) which allows the use of “work-sampling” techniques, a sampling methodology widely used for the study of work activity [5], also in the field of ergonomics [6].

The risk assessment process in a multi-site reality can logically be divided into two subgroups:
-Risks deriving from site-specific conditions (i.e., the work premises and the systems present in them);-Risks deriving from site-independent conditions (i.e., the use of particular equipment or standardized working methods).

In the first case, the assessment of the risks deriving from site-specific conditions (i.e., fire hazards) must be based on an analytical approach in each company site, since there is no way to ensure the completeness of the assessment otherwise.

When evaluating the risks deriving from site-independent conditions, the assessment of a representative condition should satisfy the whole multi-site reality (producing acceptable margins of error) and the risk calculated for a specific company could be extended to the others. For example, the risks (cuts and biomechanical factors) associated with a slicer placed on a standard-height counter in a store of a large-scale supermarket company will be identical regardless of the number of available slicers in a store or the placement of the same machine in one store or another. 

The choice of adequate company locations for risk assessment could be randomly extracted from several company sites, but only for locations with similar working activities and characteristics.

For companies showing heterogeneous activities and dimensions in the locations, a marked improvement in sampling efficiency can be obtained with the statistical technique of cluster analysis, with which a starting population is divided down into homogeneous groups: after this, random sampling is carried out within each cluster, in proportion to its number. The result is a random sample that adequately respects the typology of the entire starting population and therefore provides sufficient estimation accuracy with lower numbers than pure random sampling.

### 1.2. Preliminary Analysis: Definition of the Evaluation Methodology

Another element to be defined in advance is the methodology to be used for the technical analysis of biomechanical factors, which may vary according to the specific operational context. All the selected methods, proposed for risk evaluation, have a solid scientific background published in peer-reviewed journals or recognized books in the occupational setting; furthermore, data on repeatability, validity, and applicability in field studies are available. Currently, three levels of progressive complexity may be distinguished, depending on the specific purposes.

Preliminarily, it is necessary to identify a type of activity for which, from a scientific point of view, exposure to biomechanical factors can be considered negligible, as it is substantially overlapping with that associated with common life activities (washing, dressing, housekeeping, taking care of family members, buying the necessary goods, moving, traveling, and so on). From a practical point of view, this activity can be identified as a common clerical activity (where no special conditions are present).

A first level of evaluation (“screening”) aims to identify the presence of biomechanical factors which, due to the characteristics, intensity, and/or duration of exposure, deserve a deeper evaluation. The assessment tool chosen for this type of investigation must be complete, i.e., able to analyze the presence of all biomechanical potential hazards in each body area, considering the duration and frequency of exposure. The first-level assessment efficiently identifies the situations in which an additional assessment is indicated, thus being able to distinguish the different tasks into “adequate” (i.e., they do not require further investigation) and “potentially inadequate” (i.e., they require an additional assessment to better define the level of associated biomechanical load).

A second level of evaluation involves the use of methods, generally observational, which are necessarily different from each other: some methods focus on specific anatomical sites or risk factors, while others propose risk indices combining several factors such as repetitiveness, strength, postures, loads handled, and environmental dimensions. The choice of the most appropriate method needs to consider the specific conditions of the case to be examined, emerged from a first-level analysis or previous evaluations. The methods used must have a solid reference base supported by scientific evidence and evaluated both in terms of reproducibility and validity. Methods that have demonstrated good reproducibility and a good ability to identify exposures capable of effectively increasing the risk of developing disorders of the musculoskeletal system warrant the identification of situations worthy of preventive or corrective interventions.

A specific aspect to consider is that of ISO standards; these are voluntary standards, which can be adopted by an organization also to evaluate biomechanical factors, but cannot be considered “best practice” from a scientific point of view, as recently authoritatively stated [7].

If the second-level analysis proves insufficient, or if specific problems arise requiring the measurement of complex parameters (for example, the measurement of physiological parameters such as heart rate, muscle bioelectrical activity, or oxygen consumption, movement analysis, or biomechanical and strength prediction models), it may be necessary to resort to a third level of evaluation, generally carried out utilizing instrumental measures (this could encompass special equipment and experts in this field).

## 2. Description of Methods for the Analysis of Biomechanical Factors

This section reports methods that can be adopted for the evaluation (first- and second-level) of biomechanical factors, which meet the criteria listed above (validity, completeness, and adequate use of resources).

### 2.1. First-Level Analysis

The preliminary “screening” of biomechanical factors could be performed using two checklists from the Department of Labor and Industry of the State of Washington, USA, which have been scientifically evaluated and appropriately used in the field.

These tools are aimed at reducing workers’ exposure to biomechanical factors that can cause (or increase the frequency of) musculoskeletal disorders. The checklists combine simplicity and immediacy of use with completeness and sufficient methodological rigor: in fact, they take into consideration various biomechanical factors (posture, repetitiveness, force, vibrations, and the manual handling of loads) and their duration and frequency [2,8].

According to this method, the analysis of biomechanical factors is a two-step process. First of all, the existence of “caution-zone” jobs (https://lni.wa.gov/safety-health/_docs/CautionZoneJobsChecklist.pdf, accessed on 3 September 2024) is assessed, i.e., tasks or factors in which certain characteristics occur, highlighting potential biomechanical overloads for body areas (upper limbs, cervical spine, lumbar spine, and lower limbs). The caution-zone checklist is then supplemented by the “hazard-zone” checklist (https://lni.wa.gov/safety-health/_docs/HazardZoneChecklist.pdf, accessed on 3 September 2024), identifying the work tasks likely to pose a risk for employees, for which it is therefore appropriate to resort to further assessments. An advantage of the hazard-zone checklist is the indication of exposure time limits to specific factors, thus being able to act as a guide towards risk reduction measures.

In a cross-sectional study conducted in the solid waste collection and management sector, an association with musculoskeletal disorders was observed (in particular, the congruence between the disease/disorder detected, and the risk factor assessed by the checklist as “positive”). The tool showed high sensitivity and moderate specificity [9].

This method is also part of a standard for preventive impact assessment, which has demonstrated its effectiveness in the field [10].

The checklist related to load manual handling has also been applied to evaluate lifting tasks in a large-scale distribution context: the correspondence with the results provided by other methods (including the NIOSH equation) has been judged to be moderate [2,8].

Finally, the repeatability between observations by different operators was investigated and was judged to be moderate [11].

More recently, a study by Sala and colleagues [12] confirmed the good screening power of the Washington State Caution Zone Checklist, when used as a preliminary risk-assessment method for the upper limbs.

However, the manual pushing and pulling of loads are not taken into account in the checklist and need to be evaluated using force measurements: only at the screening level, data from the scientific literature could be used [13]. For example, pushing a four-wheeled trolley (with a total mass of 130 kg) on a flat surface for 10 m involves the development of forces (initial and maintenance) that are compatible with the values for the female population derived from the psychophysical studies of Snook and Ciriello, used for evaluating the horizontal manual handling of loads [13].

### 2.2. Second-Level Analysis

For the second-level assessment of biomechanical risk, several tools are available, consisting of both voluntary technical standards (e.g., ISO 11228 standards parts 1, 2, and 3 [14,15,16]) and methods published in the international scientific literature: a set of techniques allowing a sufficiently complete assessment of biomechanical factors is briefly illustrated below.

#### 2.2.1. Vertical Manual Handling of Loads (Lifting and/or Lowering Actions)

For the second-level evaluation of lifting (and/or lowering) actions, the most internationally known method is the equation of the National Institute for Occupational Safety and Health of the USA for the calculation of the Lifting Index (LI).

In 1981, the National Institute for Occupational Safety and Health (NIOSH) proposed a method for assessing the biomechanical risk associated with repeated lifting. This equation was later modified in 1991 and published in 1993 and 1994 to broaden its applicability to more complex lifting tasks (not only those performed in the sagittal plane) [17,18,19,20].

The NIOSH model starts from a reference mass of 23 kg, considered acceptable for men and women within “ideal” lifting conditions (straight back, from knuckle height to shoulder height, i.e., mainly with action on the arms, while the muscles of the trunk and lower limbs essentially work to keep the body stable in the position), which is (de)multiplicated according to coefficients that depend on the temporal and spatial characteristics of the lift and varies from 0 to 1 (optimal conditions). When the task under evaluation involves conditions deviating from the optimum, the relative coefficient takes smaller values the greater the distance from the so-called “ideal” conditions.

The value of the final “recommended” weight is then compared with the weight lifted in the work activity: if the weight to be lifted is lower than the recommended one, the task is acceptable.

The NIOSH model is “well documented and tested in numerous laboratory studies” and associated with a “solid background based on scientific studies” [2]. The revised equation is based on three criteria deriving from biomechanical, physiological, and psychophysical scientific studies. In particular:
-The biomechanical criterion is oriented towards limiting the load on the lumbosacral spine (most important in the occasional lifting of very heavy loads);-The physiological criterion is oriented towards limiting the overall work (oxygen consumption) associated with repeated lifting tasks;-The psychophysical criterion is instead oriented towards limiting the load based on the perception of the workers’ lifting capacity (subjective assessment).

In many work contexts, workers must perform different lifting tasks: in some circumstances, it is possible to calculate (as indicated by NIOSH in 1994) a Composite Lifting Index (CLI) which would be useful for evaluating multiple lifting tasks [17]. A classic example of multiple lifting tasks is represented by the activity of breaking down a pallet containing identical objects (the variable “height of the hands from the ground” changes lift after lift). In a prospective study, it was observed that CLI values greater than 2 would be predictive of self-reported back pain [21].

Several (cross-sectional) studies have evaluated the association between LI and low back pain. In particular, Marras et al. proved the ability of the method to correctly identify jobs with a high, medium, and low risk of low back pain: the tool showed high sensitivity and moderate specificity (meaning that a tendency of the method to overestimate the risk was observed, classifying more tasks at high risk) [22].

In another study [23], it was observed that the form of the dose–response relationship between the increase in the LI and the increase in the prevalence of low back pain is not linear: for an LI higher than 3, there is a reduction in the prevalence of symptoms. The authors themselves cautiously conclude that longitudinal studies are needed to characterize the dose–response relationship.

The results of the studies available to date on the association between LI and/or CLI and the (self-reported) risk of low back pain do not appear consistent: a longitudinal survey published subsequently showed that the risk of developing low back pain increased for each unit increase in peak LI and peak CLI. The peak LI suggested a continuous increase in risk, as the index itself increased, while the trend did not occur for the peak CLI [24]. The same authors studied the relationship between the LI and CLI evaluation and the risk of consulting health care providers for low back pain. It turned out that the risk increased for each unit increase in peak LI and peak CLI. The peak CLI suggested a continuous increase in risk as the index increased: this condition did not occur for peak LI [24].

In another study, peak LI and CLI were associated with the risk of low back pain [25]: however, in a subsequent review, the authors concluded that further studies are still needed for a validation of the estimated risk function with LI or CLI [26].

The NIOSH method for the evaluation of lifting tasks has been transposed, with modifications, into the ISO 11228-1 standard [14]. The ISO standard is conceptually similar to the NIOSH equation, since it leads to the determination of a “recommended weight” through an equation that, starting from an initial reference mass that can be lifted in optimal conditions (mref), introduces unfavorable elements with (de)multiplication factors (using the coefficients of the NIOSH method) [27].

As recently stated in the scientific literature, ISO standards in the ergonomic field do not necessarily comply with scientific criteria [7]. Since the 11228-1 ISO standard is a transposition of the NIOSH model, for practical purposes, they can be considered interchangeable, with the only difference being the reference mass: the NIOSH model indicates 23 kg for men and women alike, while the ISO standard (2021 revision) suggests a variety of reference masses, from 15 to 25 kg [14].

Given the different biomechanical capacities of men and women, it has become customary to evaluate the tasks of the vertical manual handling of loads using different reference masses by sex and age: in particular, according to the ISO 11228-1 standard, 25 kg is considered protective for 95% to 99% of the adult male working population, while 20 kg is considered protective for 85% of the female population and 99% of the general male working population. Therein lies the major difference between the ISO 11228-1 standard and the method recommended by NIOSH, which instead uses a single reference mass (load constant) of 23 kg (51 pounds), stating that “Although the load constant of 23 kg was based on a weight limit acceptable for 75% of female workers, the recommended weight limits are acceptable for at least 90% of female workers, when the revised load constant is applied in the lifting equation” [18]. Moreover, the reference masses indicated in the ISO standard are a collection of data extrapolated from heterogeneous sources whose validity cannot be independently verified. But the major problem posed by the ISO 11228-1 standard lies in using the multipliers of the NIOSH equation by proposing (also) different theoretical reference masses: the authors of the standard do not provide any evidence, theoretical or empirical, that justifies the validity of using these multipliers with reference masses that differ from those of the NIOSH model.

Consistent studies on the manual lifting capacity of loads, however, exist: for example, normative values have been published for the lifting capacity of 4443 men (41%) and women (59%) aged 15 to 65 years, based on the EPIC Lift Capacity test [28]. It is worthwhile to compare the results of this study as these values could be used as a control of other models that examine the geometries of the lift, such as the NIOSH method.

As an example, the comparison between the value of the recommended weight limit (RWL) for the three types of lifting (waist to shoulder lift, floor to waist lift, and floor to shoulder lift), obtained through the application of the NIOSH method (weight constant of 23 kg) and the normative values (10% least performing of the sample) of lifting capacity (in kg), by age and gender, based on the EPIC Lift Capacity test, give mixed results.

As far as males are concerned, the NIOSH values are comparable with the normative values relating to the 10% least performing of subjects between 50 and 59 years of age, while they are higher than the normative values for females. From this comparison, the values calculated with the NIOSH method would seem to be overprotective, as Potvin already suggested in 2014 [29], for younger underperforming males, adequate for older underperforming males, but not sufficiently protective for less performing females, in particular for older ones. However, for 75% of women, NIOSH values would be lower than those based on the EPIC Lift Capacity test. Furthermore, the recommended weight values for the NIOSH method decrease in the waist to shoulder → floor to waist → floor to shoulder direction, while according to the EPIC test, the lifting with the highest normative value is the floor to waist one, followed by waist to shoulder and floor to shoulder. According to the EPIC values, lifting capacity is globally maintained in both sexes up to the age of 49 and (moderately) decreases only in the following decade.

A separate situation, which requires specific evaluation, is that of “infrequent lifting” or, for example, lifting/lowering tasks with limited durations (i.e., a few minutes) or occurring sporadically during the work shift, or not even daily. In these cases, the NIOSH method or all its derivatives (e.g., the aforementioned ISO standard) tend to overestimate the biomechanical load because they introduce variables related to physiological or psychophysical criteria that are irrelevant for short durations: all tasks with these characteristics can be better evaluated using biomechanical models such as, for example, the 3D Static Strength Prediction Program of the Center for Ergonomics of the University of Michigan (USA). The model has been applied to evaluate load storage operations in a supermarket chain [8]: for lifting tasks carried out close to ground level (in particular, at 25 cm height from the ground level), the results showed lifting indexes comparable to those obtained by other methods (e.g., the NIOSH equation). In the evaluation of tasks in which other factors (such as the frequency and duration of operations) are more important, the lifting indexes were found to be on average lower than those obtained by applying the NIOSH method.

##### Representation of the Variability of Lifting Tasks

Numerical indexes may not always provide an adequate representation of the overall manual handling of loads carried out by a person on a work shift, if this is, as a rule, variable: in fact, many work tasks involve highly variable lifting for both the mass handled and the conditions in which the lifting itself is carried out. However, even in the last edition of the NIOSH Application Manual (revised in September 2021) [30], the examples reported are “introductory”, as they are meant to instruct neophytes into the field. Here, we want to provide supplementary recommendations to assess variable lifting tasks (i.e., different masses and/or geometries) performed by workers.

A schematic step-by-step summary of the proposed biomechanical risk assessment for manual material handling is reported in Figure 1.

##### Representation of Variability Due to Masses

Since many work environments are characterized by tasks that involve highly variable mass moved, it is necessary to represent this variability during the work shift with an image (albeit approximate) of the total load. It is therefore useful to identify an average mass lifted and the typical conditions of the lifts, since it is more likely that a worker is engaged in the most frequent operations during the whole working day. To have a proper representation of the biomechanical exposure of the whole job, handling operations that are at the highest end of the mass distribution should also be assessed according to their frequency.

If we consider (for example) the NIOSH method, the size of the mass handled and the horizontal distance of this from the body are the variables that most affect the calculation of the Lifting Index: therefore, it is evident that identifying a range of masses representative of all those handled is useful because the masses lower than the representative range are included in the evaluation that uses this value. The larger ones can then be considered using the specific frequency with which the movement takes place.

Therefore, the evaluation of lifting operations should be carried out considering the results of a statistical analysis of all loads handled manually in a work shift. If the distribution of activities among different work shifts is not homogeneous, the number of shifts considered for the calculation of the representative statistical parameters must be indicative of this variability.

To evaluate whether the distribution of manually handled weights can be approximated to the normal (Gaussian) one and therefore be able to consider the mean as a representative measure of the population, the kurtosis and symmetry test and the Shapiro–Wilk test can be used. In the absence of a normal distribution, the data are tested for possible mathematical transformations that may eventually bring the data closer to the Gaussian curve. If the latter solution is not satisfactory, the analysis of the fundamental descriptive statistics should be used, trying to find a significant value from the balance between the mean, median, and quartiles of distribution. 

Basically, we recommend calculating the weighted average of the mass distribution, which can be used as a single-load value considering the “whole activity effort” and can be employed for assessing the biomechanical risk (i.e., inserted in the NIOSH equation). However, when the weight distribution could be divided into two or more populations, given the non-negligible frequency of other masses, it may be appropriate to calculate the weighted averages of the subpopulations and then proceed with similar but separate analyses.

##### Representation of Variability Due to Geometries

Many work contexts are characterized by tasks that involve lifting actions that are rather variable in terms of the geometry of the lifts themselves. Therefore, it is necessary, as for the masses, to extract from the variability of the lifts carried out by a person in a work shift an image (albeit approximate) of the geometries occurring most frequently (typical geometry).

Typical geometry is defined by the inspection of the work activity to identify which are the most recurrent picking and depositing points. If a single sampling point and a deposition point can be identified, the evaluation can be carried out using the NIOSH method described above (Lifting Index). On the other hand, assuming that the objects to be lifted are handled at different heights and/or horizontal distances, as, for example, in the activity of breaking down a pallet and positioning the load on a bench at a fixed height, it becomes necessary to use a summary index of the lifting activity that adequately and conservatively represents the distribution of all the lifting indices that can be calculated for a given task (i.e., an index for each lifting action).

##### Job Rotation and Workstation Variability

Task rotation is often used as an organizational measure to reduce occupational exposure to biomechanical risk factors. Some methods exist to calculate the relative physical demands for manual lifting jobs where workers perform a sequence of lifts in a workstation and then move to another one to complete a different series of lifting operations [31]. There are also jobs characterized by individual lifting actions that vary from lift to lift due to the task requirements [32]. The scientific base for the calculation of these metrics is similar to that for the LI (or CLI) procedure originally published by NIOSH; however, as stated by the authors, these methods still have to be validated [32]. Moreover, the application of these tools requires time as they are not “pen and paper”.

##### Summary Lifting Indexes

In many situations, the manual lifting of loads involves different masses and geometries that cannot be reduced to a single lifting index. However, it is also essential to have a synthetic representation of the activity to understand whether it can be considered acceptable or not.

For the representation of different lifts, a systematic procedure can be followed:(1)Identify and exclude from the lifting evaluation loads which, due to their mass, do not reasonably constitute a risk to workers, regardless of the geometry of the lift. For example, considering that an upper limb makes up about 3% of body mass (both, therefore, 6%) and assuming that the average weight of a woman is 66 kg and that of a man 81 kg (values calculated for the Italian population by Donfrancesco C, et al. [33]; values for other countries may vary), the weight of the two arms in a woman is about 4 kg and that in a man about 5 kg. Assuming that lifting a load of a weight similar to that of one’s arms does not constitute a significant load from a biomechanical point of view (for non-awkward postures and if the frequency is not too high), masses of this order can be subject to risk assessment from manual handling only if the frequency with which they are lifted is such that the risk may exceed the physiological criterion (oxygen consumption) underlying the NIOSH method (the ISO 11228-1 standard, for example, always excludes from the evaluation masses less than 3 kg and, under specific conditions, those between 3 and 5 kg).(2)Evaluate the lifting of loads above the minimum limit where they are different in mass and geometry; the calculation of the Composite Lifting Index (CLI) can be used, as reported in the NIOSH publication [30], even though it has to be kept in mind that CLI has not yet received robust empirical validation.

In addition, the inconsiderate use of CLI where the variations concern only the mass and geometries of the lifts, and not the frequency, leads to paradoxical results in which CLI may give values well higher than the LI of each task separately evaluated.

Since it is not validated by adequate epidemiological studies, the interpretation of CLI remains associated with high uncertainty. As stated in a recent article [34], the set of (limited) available evidence suggests that a CLI value higher than 2 is an indicator of increased risk of back pain, while at lower values this risk is not evident.

##### Representation of Lifting Indexes for Different Reference Masses Based on Gender or Age

It is possible to evaluate vertical manual handling according to the ISO 11228-1 standard, using 25 kg as reference mass for men (considered protective for 95–99% of the adult male working population) and 15 kg for women (considered protective for 90% of the general female working population).

However, in companies where the workforce mainly consists of one gender (e.g., men in construction or women in the clothing industry), to facilitate data readability, it is preferable to choose only the reference mass of the most represented sex.

For some subpopulations, however, it may be necessary to calculate LI for different masses by multiplying the LI calculated for a reference mass (e.g., 25 kg) with the ratio between the mass used to calculate the first index and the mass used to calculate the new one. Thus, an LI of 1 for a reference mass of 25 kg becomes 1.25 for a reference mass of 20 kg (1 × 25/20).

However, the lack of scientific evidence supporting the validity of the NIOSH equation should be borne in mind when using different reference masses in the original model.

#### 2.2.2. Horizontal Manual Handling of Loads (Pulling and/or Pushing, Transport)

Pushing and pulling loads cannot be evaluated by observation, but force measurements are required in real-life situations. The evaluation of the acceptability of such actions is carried out based on data from psychophysical studies [35,36,37]. The Liberty Mutual Manual Materials Handling Tables, which represent the most recent update of the studies mentioned above, are available at https://libertymmhtables.libertymutual.com/tasks/pushing/, accessed on 4 September 2024.

The psychophysical criterion is based on the opinions of volunteers subjected to laboratory experiments, in which they had to lift/lower, push/pull, or carry loads of (to them) unknown weights for limited periods (up to 4 h). The subjects were asked to work “to the best of their ability, without hurting themselves or becoming too tired, fatigued, hot or short of breath” [36]. The ability of the method (with particular reference to the activities of the vertical handling of loads) to correctly identify jobs with high, medium, and low risk of the onset of back pain was studied [22]: the psychophysical tables showed moderate sensitivity and high specificity.

The ISO 11228-2 standard also refers (method 1) to the data obtained from the psychophysical experimental studies mentioned above [36] to evaluate the acceptability of the values of pushing and pulling force (initial and sustained) [15].

In addition, biomechanical methods can be used to calculate the vertical and horizontal forces that develop at the level of the trunk during pulling and/or pushing actions. These methods integrate, in the final evaluation, the contribution of factors such as posture, anthropometric characteristics, and the sex of the subject performing the action. Among these tools, an example is represented by the 3D Static Strength Prediction Program of the Center for Ergonomics of the University of Michigan (USA) which also allows researchers to calculate the coefficient of friction and to evaluate the acceptability of the task, aiming to prevent any accidents at work (falls/slips). This assessment method has been incorporated into the ISO 11228-2 standard (method 2: “specialized risk estimation and risk assessment approach”).

The manual “carrying” of loads is generally infrequently performed in the workplace, as it is carried out much more efficiently by machines and, therefore, it rarely reaches a significant level from the point of view of health risk. Indications on the reference values relating to the manual “carrying” of loads are, however, included in the tables of psychophysical data described above [36]. These tables have been incorporated into the ISO 11228-1 standard, as reference values to assess the acceptability of the transport action (as, in psychophysical studies, it is necessary to determine the frequency of transport, the length of the route, and the extent of the load transported).

## 3. Conclusions

The widespread diffusion of biomechanical risk factors together with the high prevalence of musculoskeletal disorders in the world population have prompted, in the past 40 years, the publication in the scientific literature of several methods for the assessment of biomechanical exposure.

Many of these methods have had little application and lack validation of their predictive claims through longitudinal epidemiologic studies. The methods presented in this two-part guide stand out as the ones for which more data regarding their validity are reported. The use of these tools, however, needs a thorough understanding of the principles on which they are based and the limitations concerning applications and predictive value.

Specifically, here, we underline the importance of a two-step approach for biomechanical risk factor evaluation: a screening analysis and, when needed, a more in-depth risk assessment.

Among the methods to assess vertical manual material handling operations, we recommend using the revised NIOSH equation for which more data are reported, thus remaining the most suitable method for assessing biomechanical risk in lifting tasks. Notably, in many studies, it is used as a reference method with which to compare the results of other tools [38,39]. Moreover, the revised NIOSH equation, supported by biomechanical, physiological, and psychophysical criteria, provides very conservative results and is therefore very protective to ensure the health of workers.

However, to advance the current knowledge about biomechanical risk factors (to the level, for example, reached for several physical and chemical agents), we need high-quality cohort studies in which the methods herein presented are applied together with objective instrumental exposure measurements; health outcome needs to be assessed objectively, with the best available appropriate diagnostic techniques (electromyography, diagnostic imaging, and others), and complete control of confounders and possible sources of bias has to be assured. These studies would provide accurate (enough) information to establish the predictive validity of any methods of risk assessment.

## Figures and Tables

**Figure 1 life-14-01398-f001:**
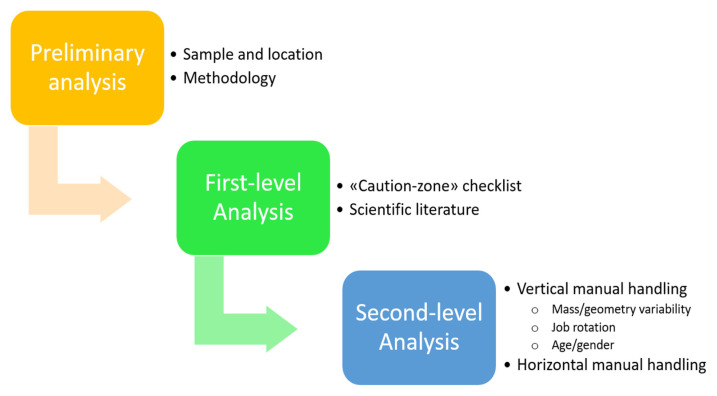
Schematic representation of the biomechanical risk assessment evaluation procedure for manual material handling.

## Data Availability

No new data were created or analyzed in this study.

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
