# Peer review of "Criteria for Assessing Exposure to Biomechanical Risk Factors: A Research-to-Practice Guide—Part 1: General Issues and Manual Material Handling"

_life, 2024, doi:10.3390/life14111398_

Round 1
Reviewer 1 Report
Comments and Suggestions for Authors
The article focuses on manual handling tasks, trying to analyze some related aspects in a more in-depth way. However, it omits important aspects and seems to not address the main aspect related to the presented objective. The title suggests that this is the first part. I am not familiar with the second part. Perhaps the article would benefit from combining both parts. The main shortcomings of the article are listed below:
The title and the objective of the study indicate that the article refers to general exposure to biomechanical factors related to different tasks, but the content of the article focuses only on lifting and pulling/pushing. I suggest to modify both the title and the objective.
The publications cited in the article are mostly relatively old, and the NIOSH equation is widely known and published in various contexts. The authors did not emphasize what is novelty in their article and why it is worth publishing. This is the first part, I assume the introductory part. Perhaps it would be better to combine both parts to get a specific result.
When the goal is stated as: “this work aims to bridge the gap between research and practice, presenting a set of criteria based on the most valid and up-to-date scientific evidence.”, I would expect the latest scientific evidence to be presented and combined with existing methods, and clear recommendations for practitioners as a summary. Unfortunately, scientific evidences are sparce and a there is a lack of recommendations or summary.
Why Authors included the subsection “1.1 Preliminary analysis: definition of the field of evaluation” if title already says that they would focus on biomechanical factors.
Subsection "2.1 First-level analysis" - is it necessary to apply any checklist if there is manual handling? Or is a risk assessment not necessary in such a case.
Subsection "2.2 Second-level analysis" is the main part of the article. However, all the information that could be interesting in the context of the article's purpose is written in a rather obscure way. One would expect it to be written in a more informative way including clear recomendations.
The subsection “2.2.1.1.1 Representation of variability due to masses” would be closest to the study objectives if it were presented in a more consistent way, for example the presented values ​​were compared in tables. Clear conclusions/recommendations from this analysis would also be expected.
A very common way to reduce exposure is to jobs rotation. Situations where the employee is lifting or pushing all day long are very rare. One would expect that the job rotation aspect would also be analyzed.
The conclusions section should relate strictly to the content of the article concluding it in a short and consisted way. Hear the conclusions section introduce a new issue, which is the description of experimental methods for assessing load. A description of such methods, together with their advantages and disadvantages, would rather be expected as part of the article.
Author Response
The article focuses on manual handling tasks, trying to analyze some related aspects in a more in-depth way. However, it omits important aspects and seems to not address the main aspect related to the presented objective. The title suggests that this is the first part. I am not familiar with the second part. Perhaps the article would benefit from combining both parts. The main shortcomings of the article are listed below:
The title and the objective of the study indicate that the article refers to general exposure to biomechanical factors related to different tasks, but the content of the article focuses only on lifting and pulling/pushing. I suggest to modify both the title and the objective.
A: We thank th Reviewer for the comments. The manuscript was conceived as a single article with the complete evaluation of biomechanical factors, but the Editor suggested to split our study in two manuscripts, thus having more words for better detail and describe the work activity assessment. As you suggest, we tried to be clearer by changing the title in “Criteria for assessing exposure to biomechanical risk factors: a research-to-practice guide. Part 1: General issues and Manual Material Handling”.
In the second part, we comment and discuss some of the most documented (and used) methods for the assessment of biomechanical exposure of the upper limb in the workplace. Specifically, for the second-level analysis of manual activities requiring speed, continuity of movement and use of force, the method having more validation data is the one proposed by the American Conference of Governmental Industrial Hygienist (ACGIH®). We also present some risk assessment methods for the evaluation of shoulder posture, upper and lower limb fatigue. Finally, we present criteria to proportionate risk assessment to the working duration in part-time jobs.
The publications cited in the article are mostly relatively old, and the NIOSH equation is widely known and published in various contexts. The authors did not emphasize what is novelty in their article and why it is worth publishing. This is the first part, I assume the introductory part. Perhaps it would be better to combine both parts to get a specific result.
A: As the Reviewer observed and as we explained in the above response, our study is composed of two parts (following the Editor’s suggestion) named “Criteria for assessing exposure to biomechanical risk factors: a research-to-practice guide. Part 1: General issues and Manual Material Handling” and “Criteria for assessing exposure to biomechanical risk factors: a research-to-practice guide. Part 2: Upper and Lower Limb”. In the Abstract we explained this division.
To highlight the novelty of our article, we added in the Introduction section the following sentence:
“Finally, we aim to provide some practical recommendations to collect, manage and analyze data.”
And we added a new phrase in the paragraph 2.2.1.1 Representation of the variability of lifting tasks:
“However, even in the last edition of the NIOSH Application Manual (Revised in September 2021) the examples reported are “introductory”, as they are meant to instruct neophyte into the field. Here, we want to provide supplementary recommendations to assess variable lifting tasks (i.e. different masses and/or geometries) performed by workers.”
When the goal is stated as: “this work aims to bridge the gap between research and practice, presenting a set of criteria based on the most valid and up-to-date scientific evidence.”, I would expect the latest scientific evidence to be presented and combined with existing methods, and clear recommendations for practitioners as a summary. Unfortunately, scientific evidences are sparce and a there is a lack of recommendations or summary.
A: We thank the Reviewer for the suggestion. As this was not a comprehensive review, we mainly focused on the methods for which more validated data were available and tried to highlight their use on the field, also for variable tasks/activities.
Moreover, we have enriched the bibliography with more recent studies and added in the Conclusion section a final recommendation on NIOSH method:
“Specifically, here we underline the importance of a 2-step approach for biome-chanical risk factor evaluation: a screening analysis and, when needed, a more in-depth risk assessment.
Among the methods to assess vertical manual material handling operations, we recommend using the revised NIOSH equation for which more data are reported, thus remaining the most suitable for assessing biomechanical risk in lifting tasks. Notably, in many studies it is used as a reference method with which to compare the results of other tools. Moreover, the revised NIOSH equation, being supported by biomechanical, physiological and psychophysical criteria, provides very conservative results and is therefore very protective to ensure the health of workers “
Why Authors included the subsection “1.1 Preliminary analysis: definition of the field of evaluation” if title already says that they would focus on biomechanical factors.
A: The Reviewer is right that the title of this subsection is misleading. We wanted to highlight that the first step is to define the location and sample that needs to be evaluated, within the biomechanical risk field. Thus, we changed the subsection title into:
“1.1 Preliminary analysis: selection of a representative sample and location for the assessment ”
Subsection "2.1 First-level analysis" - is it necessary to apply any checklist if there is manual handling? Or is a risk assessment not necessary in such a case.
A: The checklist is always needed as first level analysis, thus including manual material handling (present in the subsection “Heavy, Frequent or Awkward Lifting” in the checklist), as it helps evaluating the activities for which the second level assessment would be necessary.
We added a more recent literature reference confirming the good screening power of the Washington State checklist and a sentence to explain it:
“More recently, a study by Sala and colleagues confirmed the good screening power of the Washington State Caution Zone Checklist, when used as a preliminary risk-assessment method for the upper limbs.”
In the case of horizontal handling operations (manual pushing or pulling), which is not present in the checklist, we proposed to use some data from literature as a screening tool (rows 203-209). To better clarify this, we rephrased the sentence at the end of the paragraph into:
“However, manual pushing and pulling of loads are not taken into account in the check-list and need to be evaluated using force measurements: only at the screening level, data from scientific literature could be used.”
Subsection "2.2 Second-level analysis" is the main part of the article. However, all the information that could be interesting in the context of the article's purpose is written in a rather obscure way. One would expect it to be written in a more informative way including clear recomendations.
A: Thank you very much for this suggestion. We provided some final recommendations throughout the while manuscript, answering and complying also with the other comments and suggestions. Furthermore, we added the following sentence in the Conclusion:
" Specifically, here we underline the importance of a 2-step approach for biome-chanical risk factor evaluation: a screening analysis and, when needed, a more in-depth risk assessment.
Among the methods to assess vertical manual material handling operations, we recommend using the revised NIOSH equation for which more data are reported, thus remaining the most suitable for assessing biomechanical risk in lifting tasks. Notably, in many studies it is used as a reference method with which to compare the results of other tools. Moreover, the revised NIOSH equation, being supported by bio-mechanical, physiological and psychophysical criteria, provides very conservative re-sults and is therefore very protective to ensure the health of workers. "
The subsection “2.2.1.1.1 Representation of variability due to masses” would be closest to the study objectives if it were presented in a more consistent way, for example the presented values ​​were compared in tables. Clear conclusions/recommendations from this analysis would also be expected.
A: The Reviewer is right and we added a final suggestion for the reader aiming at reporting a “take-home message” from this section, as also was previously mentioned that we are missing some recommendations along the text:
“Basically, we recommend calculating the weighted average of the mass distribu-tion, which can be used as single load value considering the “whole activity effort” and can be employed for assessing the biomechanical risk (i.e. inserted in the NIOSH equation). However, when the weight distribution could be divided into two or more populations, given the non-negligible frequency of other masses, it may be appropriate to calculate the weighted averages of the subpopulations, then proceeding with similar but separate analyses.”
A very common way to reduce exposure is to jobs rotation. Situations where the employee is lifting or pushing all day long are very rare. One would expect that the job rotation aspect would also be analyzed.
A: We agree with the Reviewer’s comment as tasks rotation is a method often used to reduce occupational exposure to biomechanical risk factors. For this reason, we have treated this problem in the second part of the manuscript, but as he/she suggested we also added some details into a specific section named “Job rotation and workstation variability” in this first part:
“Tasks rotation is often used as an organizational measure to reduce occupational exposure to biomechanical risk factors. Some methods exist to calculate the relative physical demands for manual lifting jobs where workers perform a sequence of lifts in a workstation and then move to another one to complete a different series of lifting operations. There are also jobs characterized by individual lifting actions that vary from lift to lift due to the task requirements. The scientific base for the calculation of these metrics is similar to that for the LI (or CLI) procedure originally published by NIOSH; however, as stated by the authors, these methods still have to be validated. Moreover, the application of these tools requires time as they are not “pen and paper”.
The conclusions section should relate strictly to the content of the article concluding it in a short and consisted way. Hear the conclusions section introduce a new issue, which is the description of experimental methods for assessing load. A description of such methods, together with their advantages and disadvantages, would rather be expected as part of the article.
A: We thank the reviewer for this suggestion. We addede the following paragraph, to provide some final recommendations:
"Among the methods to assess vertical manual material handling operations, we recommend using the revised NIOSH equation for which more data are reported, thus remaining the most suitable for assessing biomechanical risk in lifting tasks. Notably, in many studies it is used as a reference method with which to compare the results of other tools. Moreover, the revised NIOSH equation, being supported by bio-mechanical, physiological and psychophysical criteria, provides very conservative re-sults and is therefore very protective to ensure the health of workers.”
Plus, we deleted the last paragraph in the Conclusion.

Reviewer 2 Report
Comments and Suggestions for Authors
The article reviews methods for assessing risks involving manual materials handling. The aim “is to bridge the gap between research and practice, presenting a set of criteria based on the most valid and up-to-date scientific evidence”. In this, Part 1 of two, the focus is on assessment methods that have the strongest scientific foundation for manual lifting tasks.
The strength of the article is the critique of the current NIOSH Lifting Equation. Numerous published studies have reported mixed findings regarding validity of the NIOSH method. Another strength is the distinction between two stages of evaluation: a checklist type of method and a more detailed method like the NIOSH Lifting Equation.
Comment 1 concerns section 1.2 about the definition of the field of evaluation. The section speaks of starting an assessment by determining some characteristics of the business, and this is linked to instructions on using a sampling strategy to pick a representative person performing a representative task. Two examples mentioned in the paper are assessing specific tasks performed at multiple sites, like your grocery store examples. I think the paper should be more direct on the topic of extrapolating from assessments of a small sample of tasks and individuals to other tasks and persons.
Comment 2 concerns the research-to-practice aim of the paper. My take away from reading the paper about the NIOSH Lifting Equation is the paper emphasizes the research part while providing limited support for the practice part. Apparently, no attempt to validate the NIOSH Lifting Equation in practice is sufficient for the authors. There is a need to provide explicit conclusions regarding where the science is weak on achieving the research-to-practice aim of the paper. Do the authors recommend employers use the NIOSH method in their work settings?
Comment 3 concerns the presentation. I suggest developing some graphics to break up the all-text presentation.
Author Response
Reviewer #2
The article reviews methods for assessing risks involving manual materials handling. The aim “is to bridge the gap between research and practice, presenting a set of criteria based on the most valid and up-to-date scientific evidence”. In this, Part 1 of two, the focus is on assessment methods that have the strongest scientific foundation for manual lifting tasks.
The strength of the article is the critique of the current NIOSH Lifting Equation. Numerous published studies have reported mixed findings regarding validity of the NIOSH method. Another strength is the distinction between two stages of evaluation: a checklist type of method and a more detailed method like the NIOSH Lifting Equation.
Comment 1 concerns section 1.2 about the definition of the field of evaluation. The section speaks of starting an assessment by determining some characteristics of the business, and this is linked to instructions on using a sampling strategy to pick a representative person performing a representative task. Two examples mentioned in the paper are assessing specific tasks performed at multiple sites, like your grocery store examples. I think the paper should be more direct on the topic of extrapolating from assessments of a small sample of tasks and individuals to other tasks and persons.
A: To be clearer we changed the sentence in the section into:
“When evaluating the risks deriving from site-independent conditions, the assessment of a representative condition should satisfy the whole multi-site reality (producing acceptable margins of error) and the risk calculated for a specific company could be extended to the others.”
We have also deleted the sentence:
“Similarly, the handling of goods for supermarket shelfing tasks in standardized conditions (identical goods for all stores, shelves to be set up as well), could be evaluated by observing the operations from a sample of the total points of sale. “
Comment 2 concerns the research-to-practice aim of the paper. My take away from reading the paper about the NIOSH Lifting Equation is the paper emphasizes the research part while providing limited support for the practice part. Apparently, no attempt to validate the NIOSH Lifting Equation in practice is sufficient for the authors. There is a need to provide explicit conclusions regarding where the science is weak on achieving the research-to-practice aim of the paper. Do the authors recommend employers use the NIOSH method in their work settings?
A: We thank the Reviewer for the valuable comment. In our opinion the revised NIOSH equation has some limitations, specifically it lacks validation of its predictivity and, for this reason, it must be used with caution. Among the different metrics proposed over the years, the Lifting Index (and the Composite Lifting Index) is one of the tools with most data and it remains the most important and widely used to assess biomechanical risk in lifting tasks.
Moreover, the revised NIOSH equation, being based on three criteria (biomechanical, physiological and psychophysical), seems to provide very conservative results and is therefore very protective to ensure the health of workers.
Thus, we added a take-home message in the Conclusion section of our manuscript:
“Among the methods to assess vertical manual material handling operations, we recommend using the revised NIOSH equation for which more data are reported, thus remaining the most suitable for assessing biomechanical risk in lifting tasks. Notably, in many studies it is used as a reference method with which to compare the results of other tools. Moreover, the revised NIOSH equation, being supported by bio-mechanical, physiological and psychophysical criteria, provides very conservative re-sults and is therefore very protective to ensure the health of workers. “
Comment 3 concerns the presentation. I suggest developing some graphics to break up the all-text presentation.
A: According to the Reviewer suggestion, we added at the end of Section 2.2.1.1 a schematic (Figure1) that could be used to summarize the steps described in the manuscript for assessing manual material handling risk, introduced by the following sentence:
“A schematic step-by-step summary of the proposed biomechanical risk assessment for manual material handling is reported in Figure 1.”
Figure 1: Schematic representation of the biomechanical risk assessment evaluation procedure for manual material handling.

Reviewer 3 Report
Comments and Suggestions for Authors
You have conducted a study on biological risk based on environmental exposure. I think an important topic has been covered. However, in carrying out the research, this paper was written without maintaining the most important basic principle and frame. This part is similar to the abstract. The text of the paper should be divided into sections like introduction, method, results, discussion and conclusion. In addition, even in the abstract, all important key concepts should be included in the text in a larger frame than in the following section. After being well organized, I recommend you to try again.
Author Response
You have conducted a study on biological risk based on environmental exposure. I think an important topic has been covered. However, in carrying out the research, this paper was written without maintaining the most important basic principle and frame. This part is similar to the abstract. The text of the paper should be divided into sections like introduction, method, results, discussion and conclusion. In addition, even in the abstract, all important key concepts should be included in the text in a larger frame than in the following section. After being well organized, I recommend you to try again.
A: We thank the Reviewer for the suggestion, but we conducted a study on exposure to biomechanical risk factors not on biological risk. As reported in the Instruction for Authors the structure of the article can include “Abstract, Keywords, Introduction, Relevant Sections, Discussion, Conclusions, and Future Directions”: for this reason, we didn’t divide the paper into the sections “Introduction, method, results, discussion and conclusion”. However, as suggested, we added some paragraphs and sentences into the manuscript to better explain our aim and modified the title as follows “Criteria for assessing exposure to biomechanical risk factors: a research-to-practice guide. Part 1: General issues and Manual Material Handling”.

Reviewer 4 Report
Comments and Suggestions for Authors
The authors have developed a comprehensive review that aims to examine various methodologies for assessing biomechanical risk factors, with a particular focus on manual material handling. The review seeks to identify methods that have received robust validation data, such as the Revised NIOSH Lifting Equation, and discuss their applicability in diverse work environments.
However, several observations should be addressed before recommending this work for publication.
1. Could the authors provide details of the search terms utilized, as well as the databases consulted to conduct the review?
2. Could the authors include a PRISMA flowchart?
3. The authors are requested to elucidate the selection process during the literature search: Total number of articles identified, articles selected based on title and abstract screening after removal of duplicates, and articles ultimately included in the review following full-text assessment.
4. The authors could address in the introduction section a number of effective interventions for this patient population, such as therapeutic exercise and manual therapy. It is recommended that they incorporate findings from the following two recent systematic and umbrella reviews: DOI: 10.3390/medicina59020256 ; DOI: 10.1097/PHM.0000000000002239
Comments on the Quality of English LanguageNo comments
Author Response
The authors have developed a comprehensive review that aims to examine various methodologies for assessing biomechanical risk factors, with a particular focus on manual material handling. The review seeks to identify methods that have received robust validation data, such as the Revised NIOSH Lifting Equation, and discuss their applicability in diverse work environments.
However, several observations should be addressed before recommending this work for publication.
- Could the authors provide details of the search terms utilized, as well as the databases consulted to conduct the review?
A: We thank the Reviewer for the suggestion, but we have not done a systematic Review of the literature. We wanted to prepare a manual for biomechanical risk assessment reporting and commenting on the most validated methods currently employable in the field. Thus, we do not have a string related to this work.
To better clarify this point to the readers, we changed the title as follows “Criteria for assessing exposure to biomechanical risk factors: a research-to-practice guide. Part 1: General issues and Manual Material Handling” and used “guide” instead of “review” in the abstract as well.
- Could the authors include a PRISMA flowchart?
A: We did not apply a PRISMA methodology because we didn’t conduct a scoping review, systematic review and/or a meta-analysis.
- The authors are requested to elucidate the selection process during the literature search: Total number of articles identified, articles selected based on title and abstract screening after removal of duplicates, and articles ultimately included in the review following full-text assessment.
A: As answered to the first comment, we have not done a systematic Review of the literature, since we aimed to realize a manual for biomechanical risk assessment, reporting and commenting on the most validated methods currently employable. We do not have a detailed process of article inclusion/exclusion.
- The authors could address in the introduction section a number of effective interventions for this patient population, such as therapeutic exercise and manual therapy. It is recommended that they incorporate findings from the following two recent systematic and umbrella reviews: DOI: 10.3390/medicina59020256 ; DOI: 10.1097/PHM.0000000000002239
A: As answered to the first comment, we have not done a systematic or umbrella review, since we aimed to realize a manual for biomechanical risk assessment, reporting and commenting on the most validated methods currently employable.

Round 2
Reviewer 1 Report
Comments and Suggestions for Authors
I recommend to accept the paper to publication.
Author Response
We thank the Reviewer for the comment.
Reviewer 3 Report
Comments and Suggestions for Authors
It has been greatly improved than the previous version of the paper.
Thank you for your efforts!
Author Response
We thank the Reviewer for the comment.
Reviewer 4 Report
Comments and Suggestions for Authors
Dear Authors,
The clarifications provided regarding the nature of the manuscript are appreciated, and it is understood that the objective is to create a guide rather than a systematic review. The rephrasing of the title and abstract to reflect this shift from "review" to "guide" is a commendable modification, as it more accurately conveys the aim of the work to readers.
However, several suggestions are proposed to enhance the clarity and presentation of the manuscript:
1. Clarification of the Methodology: Although a systematic review was not conducted, readers would benefit from a more transparent description of the selection process for the methodologies included in the guide. It would be advantageous to elucidate whether the approach was based on a literature review, expert consensus, or another form of methodological selection. Providing clarity on this process, even if not as detailed as a systematic review, would reinforce the scientific rigor of the guide.
2. Incorporating Relevant Research: While the focus of the work is on biomechanical risk assessment, it is recommended to integrate recent findings related to effective interventions for patients exposed to biomechanical risks. As previously suggested, including references to the following studies (DOI: 10.3390/medicina59020256 and DOI: 10.1097/PHM.0000000000002239) would contextualize the clinical relevance of the methods under discussion, particularly by offering practical interventions such as therapeutic exercise and manual therapy.
3. Additional Clarification in the Introduction: To provide a more robust foundation for the guide, it is suggested to elaborate on the rationale for highlighting certain methods as the most validated or commonly used. This would further justify the focus on specific tools, such as the Revised NIOSH Lifting Equation, and elucidate their practical application in various work environments.
4. Visual Aids: Although a PRISMA flowchart is not applicable to this manuscript, it may be beneficial to include a flowchart or diagram summarizing the decision-making process for selecting and applying the different biomechanical risk assessment tools. This would enhance the accessibility of the guide for practitioners seeking to quickly comprehend the steps involved.
It is anticipated that these suggestions will contribute to improving the clarity and impact of the manuscript. Upon implementation of these revisions, it is believed that this work will provide a valuable contribution to the field of occupational health and risk assessment.
Kind regards,
Comments on the Quality of English LanguageNo comments.
Author Response
Clarification of the Methodology: Although a systematic review was not conducted, readers would benefit from a more transparent description of the selection process for the methodologies included in the guide. It would be advantageous to elucidate whether the approach was based on a literature review, expert consensus, or another form of methodological selection. Providing clarity on this process, even if not as detailed as a systematic review, would reinforce the scientific rigor of the guide.
A: The selection was based on literature review and a consensus-based process within an interdisciplinary group of experts with over 30 years of experience in research and practice in this field.
We better explained the aim of this paper rephrasing in the Introduction the following sentences: “Several methods for the assessment of the biomechanical exposure have been proposed in the scientific literature with different level of practical feasibility: this work aims to bridge the gap between research and practice, presenting a set of criteria based on the most valid and up-to-date scientific evidence. It also considers the need to use methods that do not require disproportionate technical, material, financial, and time resources.”
However, to better explain our procedure, we added at the end of the Introduction section:
“The selection process is based on literature review and expert consensus. Specifically, all the methods selected were published in peer review journal or recognized books in the occupational field and presented most data about repeatability and validity and a clear description written in English language.”
We also reaffirmed the concept at the beginning section 1.2 Preliminary analysis: definition of the evaluation methodology:
“All the selected methods, proposed for risk evaluation, have a solid scientific background published in peer review journal or recognized books in the occupational setting; furthermore data on repeatability, validity, and applicability in field studies are available.”
All the addition are highlighted in light blue.
Incorporating Relevant Research: While the focus of the work is on biomechanical risk assessment, it is recommended to integrate recent findings related to effective interventions for patients exposed to biomechanical risks. As previously suggested, including references to the following studies (DOI: 10.3390/medicina59020256 and DOI: 10.1097/PHM.0000000000002239) would contextualize the clinical relevance of the methods under discussion, particularly by offering practical interventions such as therapeutic exercise and manual therapy.
A: As we just reported previously our study is focused on occupational risk assessment methodologies and have not investigated interventions or therapy. However, we are working on another paper about interventions where we can consider your valuable suggestions and include your references.
Additional Clarification in the Introduction: To provide a more robust foundation for the guide, it is suggested to elaborate on the rationale for highlighting certain methods as the most validated or commonly used. This would further justify the focus on specific tools, such as the Revised NIOSH Lifting Equation, and elucidate their practical application in various work environments.
A: Please, refer to what we reported in the first answer.
Visual Aids: Although a PRISMA flowchart is not applicable to this manuscript, it may be beneficial to include a flowchart or diagram summarizing the decision-making process for selecting and applying the different biomechanical risk assessment tools. This would enhance the accessibility of the guide for practitioners seeking to quickly comprehend the steps involved.
A: Thank you very much for your valuable suggestion. We stressed the methods-selecting process in the Introduction and in 1.2 sections, adding the paragraphs reported in the first answer.
